# Flexible Modeling of Net Survival and Cure by AML Subtype and Age: A French Population-Based Study from FRANCIM

**DOI:** 10.3390/jcm10081657

**Published:** 2021-04-13

**Authors:** Morgane Mounier, Gaëlle Romain, Mary Callanan, Akoua Denise Alla, Olayidé Boussari, Marc Maynadié, Marc Colonna, Valérie Jooste

**Affiliations:** 1Dijon-Bourgogne University Hospital, Registre des Hémopathies Malignes de Côte d’Or, F-21000 Dijon, France; mary.callanan@chu-dijon.fr (M.C.); allaakoua@gmail.com (A.D.A.); marc.maynadie@chu-dijon.fr (M.M.); 2UMR 1231, SAPHIHR Team, INSERM, Université Bourgogne-Franche_Comté, F-21000 Dijon, France; 3LabEX LipSTIC, ANR-11-LABX-0021, F-21000 Dijon, France; Olayide.Boussari@u-bourgogne.fr; 4Dijon-Bourgogne University Hospital, Registre Bourguignon des Cancers Digestifs, F-21000 Dijon, France; Gaelle.Romain@u-bourgogne.fr (G.R.);vjooste@u-bourgogne.fr (V.J.); 5UMR 1231, EPICAD Team, INSERM, Université Bourgogne-Franche_Comté, F-21000 Dijon, France; 6Fédération Francophone de Cancérologie Digestive, Département de Méthodologie, F-21000 Dijon, France; 7Grenoble University Hospital, Registre du Cancer de l’Isère, F-38000 Grenoble, France; mcolonna.registre@wanadoo.fr

**Keywords:** acute myeloid leukemia subtype, France, population-based study, net survival, excess mortality, cure, flexible model, age disparity

## Abstract

With improvements in acute myeloid leukemia (AML) diagnosis and treatment, more patients are surviving for longer periods. A French population of 9453 AML patients aged ≥15 years diagnosed from 1995 to 2015 was studied to quantify the proportion cured (P), time to cure (TTC) and median survival of patients who are not cured (MedS). Net survival (NS) was estimated using a flexible model adjusted for age and sex in sixteen AML subtypes. When cure assumption was acceptable, the flexible cure model was used to estimate P, TTC and MedS for the uncured patients. The 5-year NS varied from 68% to 9% in men and from 77% to 11% in women in acute promyelocytic leukemia (AML-APL) and in therapy-related AML (t-AML), respectively. Major age-differenced survival was observed for patients with a diagnosis of AML with recurrent cytogenetic abnormalities. A poorer survival in younger patients was found in t-AML and AML with minimal differentiation. An atypical survival profile was found for acute myelomonocytic leukemia and AML without maturation in both sexes and for AML not otherwise specified (only for men) according to age, with a better prognosis for middle-aged compared to younger patients. Sex disparity regarding survival was observed in younger patients with t-AML diagnosed at 25 years of age (+28% at 5 years in men compared to women) and in AML with minimal differentiation (+23% at 5 years in women compared to men). All AML subtypes included an age group for which the assumption of cure was acceptable, although P varied from 90% in younger women with AML-APL to 3% in older men with acute monoblastic and monocytic leukemia. Increased P was associated with shorter TTC. A sizeable proportion of AML patients do not achieve cure, and MedS for these did not exceed 23 months. We identify AML subsets where cure assumption is negative, thus pointing to priority areas for future research efforts.

## 1. Introduction

Acute myeloid leukemia (AML) is an aggressive cancer currently classified into 23 morphology codes in the latest WHO classification of tumors of hematopoietic and lymphoid tissues [1]. The world-standardized incidence rate is 3.62 per 100,000 person-years in Europe [2]. AML is more frequent in the elderly; the median age at diagnosis is 71 years in France, but AML is diagnosed at varying incidence rates in all age categories. Among patients with AML diagnosed between 1989 and 2010, 5-year net survival was 21% in France and in Europe and decreased with age at diagnosis [3,4].

Since the year 2000, AML has been categorized into subtypes that take into account differences in clinical presentation, disease biology, prognosis and, for some, treatment guidelines [5,6]. Although dramatic improvements in AML survival have been made possible by new treatment strategies, many patients remain uncured. Indicators for optimal, risk-adapted deployment of treatment innovations in daily practice will require more clear-cut, population-based estimates of prognosis aimed particularly at AML subtypes, where treatment advances are most urgently needed. However, AML is usually studied as a single group, and data by subtype mainly come from a few specialized registries [7,8]. While survival analyses have been widely used to assess the efficacy of patient management and treatment strategies, few studies have looked at survival by subtype or cure [9,10].

The concept of statistical cure assumes that a part of the patient population will not experience mortality due to the diagnosed cancer. In other words, the mortality observed in this group of patients becomes equal to that observed in the general population. Simultaneous analysis of the proportion of “cured” patients, the time to cure and the median survival of uncured AML patients provides a more detailed assessment of the progress in diagnostic and/or therapeutic management in a given population [11]. In this setting, the main goal of this study was to estimate, by AML subtype and sex, net survival and cure indicators according to age in a French population-based data study. This was achieved by using a flexible parametric model that allows detection of complex effects between covariates of interest and estimating cure indicators.

## 2. Materials and Methods

### 2.1. Study Design and Data Collection

This retrospective population-based study included all AML cases diagnosed between 1 January 1995 and 31 December 2015 in patients aged ≥15 years and recorded in the database of the French Network of cancer registries (FRANCIM). The diagnosis of AML was defined according to the third edition of the International Classification of Diseases for Oncology (ICD-O-3) [12]. Cases diagnosed before the ICD-O-3 publication were reviewed by hematologists and recoded using the new classification. Sixteen AML groups were defined using the Haemacare proposal according to their morphology [2]: acute promyelocytic leukemia with t(15;17) (APL); AML with other recurrent cytogenetic abnormalities (AML-RCA); AML with myelodysplasia-related changes (AML-MRC); therapy-related AML (t-AML) previously treated by cytotoxic chemotherapy and/or radiation therapy, acute panmyelosis with myelofibrosis (APMF), acute biphenotypic leukemia, followed by nine AML subtypes based on the French-American-British (FAB) classification and, finally, AML not otherwise specified (AML-NOS) (Table 1). The vital status was followed over 10 years after diagnosis or up to 30 June 2018. The proportion of patients lost to follow-up was <1%.

To ensure sufficient statistical power, only AML subtypes and sex combinations for which the number of cases at diagnosis was higher than 500 or for which the number of deaths within 10 years after the diagnosis was higher than 130 were considered for survival analyses using a strategy to check for complex effects related to age. In situations where the number of deaths within 10 years after the diagnosis was between 70 and 130, an adapted strategy was made to simplify the model by adapting the number of parameters according to the number of deaths (see Appendix A). AML subtypes with less than 70 deaths in up to 10 years after diagnosis were not retained in the survival analysis. Thus, APMF, acute biphenotypic leukemia, acute basophilic leukemia, acute megakaryoblastic leukemia and myeloid sarcoma were excluded for survival analyses in both sex and acute erythroid leukemia (AML-M6) and estimations were provided only for men.

### 2.2. Statistical Analysis

#### 2.2.1. Statistical Modeling of Excess Mortality Rate and Estimation of Net Survival without Cure Assumption

Net survival at time t since diagnosis (NS(t)) was estimated using the flexible parametric model (without cure assumption) of the cumulative excess mortality rate proposed by Nelson et al. [13]. The complex effect of age at diagnosis was included in the model as a time-dependent and non-linear effect in situations where the number of deaths within the 10 years post-diagnosis was superior to 130. A separate model was fitted for each AML subtype and sex to estimate 10-year NS and excess mortality rate (EMR). Explanations of the modeling of the baseline hazard and of the complex effect of age according to the number of deaths occurring within 10 years after diagnosis are presented in the Appendix A.

#### 2.2.2. Statistical Modeling of Cure

For each AML subtype and sex, the assumption of statistical cure was accepted when the NS curve reached a plateau and the EMR approached zero within 10 years (less than 0.05). When the assumption of statistical cure was acceptable, the cumulative EMR was modeled using the flexible parametric cure model proposed by Andersson et al. [14]. The modeling of age effects was similar to that of the method described above. The cure model retained for each AML subtype and sex is presented in the Appendix A. When the cure assumption was accepted according to AML subtype, sex and age at diagnosis, three statistical cure indicators and their trend over the age at diagnosis were estimated. Firstly, the proportion of cured patients (P) was defined as the proportion of patients who will never die of their cancer. Secondly, the time to cure (TTC) was defined by Boussari et al. [15] as the delay between the diagnosis and the time t at which P(t), the probability of belonging to the cured group at time t knowing that the patient is still alive at this time [16], reached 95%.

Thirdly, we estimated the median survival (MedS) of “uncured” patients, corresponding to the time since diagnosis at which the net survival (NS) of uncured patients reached 50%.

NS estimated from both models described above (without and with cure assumption) was compared to NS obtained using the non-parametric estimator proposed by Pohar Perme [17]. EMR estimated from a model based on step function was compared to the model baseline hazard (Appendix A).

All statistical analyses were performed using STATA^TM^, version 15 (STATACorp, College Station, TX, USA).

## 3. Results

### 3.1. Characteristics of the AML Cohort

The study cohort comprised 9453 patients diagnosed with AML classified into the following subtypes: AML-NOS (2201 patients), AML with maturation (AML-M2; 1171 patients), AML-RCA (981 patients), AML-MRC (969 patients), acute myelomonocytic leukemia (AML-M4; 844 patients), AML without maturation (AML-M1; 888 patients), acute monoblastic and monocytic leukemia (AML-M5; 825 patients), t-AML (613 patients), APL (575 patients), AML-RCA (406 patients) and 961 cases for seven other AML subtypes overall (Table 1). A slightly higher proportion of men compared to women was observed in our cohort, regardless of the AML subtype (53% in the whole AML group), except for t-AML (49%) and AML-M1 (47%) (Table 1). The median age at diagnosis varied from 53 years for patients diagnosed with AML-RCA to 74 years for AML-MRC in men. In women themedian age at diagnosis varied from 52 to 76 years old, respectively, in APL and in AML-MRC (Table 2).

### 3.2. Net Survival by AML Subtype at All Ages

For the whole AML study population, except for APL and AML-RCA, the EMR was very high just after the diagnosis, which implied that NS dropped quickly during the first year and stabilized thereafter (Figure 1A.1,A.2,B.1,B.2). Survival disparities were observed between subtypes. The 5-year NS was higher in APL at all ages (68%; 95% CI, 63–74 in men; 77%; 95% CI, 72–82 in women) and in AML-RCA (56%; 95% CI, 51–62 in men; 52%; 95% CI, 47–58 in women). For the other subtypes at all ages, the 5-year NS in men ranged from 9% (95% CI, 6–13) in t-AML to 26% (95% CI, 22–29) in AML-M1, and from 11% (95% CI, 8–15) in t-AML to 29% (95% CI, 26–33) in AML-M1 in women (Figure 1A.1,A.2; Table 3). The 10-year compared to 5-year NS probability was slightly reduced for each AML subtype and in each sex.

### 3.3. Effect of Age at AML Diagnosis on Net Survival

Modeling the influence of age (Appendix A) showed a significant effect of age at diagnosis on excess mortality with an increase in mortality with age. Therefore, a better survival in younger patients (25 years old) was observed compared to older AML patients (75 years old) (Figure 1C.1,C.2,D.1,D.2 and Table 3). The largest 5-year NS difference by age was observed for patients with a diagnosis of AML-RCA, from 86% (95% CI, 78–94) to 10% (95% CI, 4–26) in men and 84% (95% CI, 74–94) to 9% (95% CI, 4–20) in women aged 25 years and 75 years, respectively. A large age difference was also observed for women with AML-MRC: 73% (95% CI, 55–97) at 25 years of age compared to 6% (95% CI, 4–9) at 75 years (Table 3). At 5 years after the diagnosis, the worst prognosis for younger patients (25 years old) was seen in men with a diagnosis of AML with minimal differentiation (AML-M0), with an NS probability of 25% (95% CI, 9–69), and in women with a diagnosis of t-AML, with an NS probability of 12% (95% CI, 3–53) (Table 3). Contrasts between sexes are mainly observed in younger patients. While NS is, overall, higher in women than in men, this trend was different in t-AML with a lower 5-year NS in women than men (12% and 40%, respectively). A large difference was also observed in AML-M0 with 5-year NS varying from 25% in men to 48% in women (Table 3).

In older patients (75 years old), diagnoses of AML-M4 and AML-M6 in men and diagnoses of AML-MRC, AML-M0, AML-M6 and AML-NOS in women presented the worst prognosis (Table 3). In the oldest patients (75 years old), poorer prognosis could be observed from 1 year, specifically in men diagnosed with AML-RCA, AML-M1, AMl-M4, AML-M5 and AML-NOS (Figure 1C.1,C.2). No major sex difference for survival was observed in older patients.

Associations between age and excess mortality differed according to AML subtype. Three situations were observed in men: (1) a non-significant non-linear effect was observed in AML-MRC, t-AML, AML-M6 and AML-M0 (NS decreased linearly with increasing age at diagnosis); (2) a significant non-linear effect in APL, AML-RCA, AML-M2, AML-M5 and AML-NOS was observed, as NS was relatively similar from 15 to 45 years of age and decreased thereafter; (3) a significant non-linear effect was observed in AML-M4 and AML-M1, as NS was the highest for middle-aged compared to both younger and older patients (Figure 1C.1,C.2,D.1,D.2; Appendix A).

Note that in women, the effect of age was similar to that observed in men, except in AML-NOS (a linear effect of age on excess mortality was observed in women compared to a non-linear effect in men).

### 3.4. Proportion of Cured Patients and Time to Cure by AML Subtype, Sex and Age

In situations where assumption of statistical cure was acceptable, cure proportion varied according to AML subtype. In patients aged 25 years, P varied in APF and t-AML, from 79% (95% CI, 68–87) to 31% (95% CI, 8–58), respectively, in men, and from 90% (95% CI, 80–95) to 10% (95% CI, 1–31), respectively, in women (Table 4). In older patients, cure assumption was accepted for a minority of AML subtypes and for relatively few of the 75-year-old patients. For these patients, P was lower than 9% in both sexes for all subtypes, except in patients with APL (*p* = 40%; 95% CI 25–52 in men and *p* = 55%; 95% CI 41–68 in women). P decreased linearly with age in AML-MRC, AML-M6, AML-M0 and AML-NOS (for women), while age had a non-linear effect on P in other AML subtypes. In men and women with AML-M1 and in men with AML-M4, P was higher in patients aged 50 years at diagnosis (47% with 95% CI 38–56; 55% with 95% CI 46–62; 46% with 95% CI 38–54, respectively) than it was in younger or older patients (Table 4). Except for t-AML and AML-M4, P was slightly higher in women than in men.

The TTC was always less than 10 years regardless of AML subtype, sex and age. For patients aged 25 years, TTC varied from 2 years (95% CI, 0–8) for APL to 8 years (95% CI, 0–17) for AML-M2 in men and from 1 year (95% CI, 0–2) for APL to 7 years (95% CI, 0–46) for t-AML in women (Figure 2C.1; Table 4). For patients aged 75 years, TTC fluctuated from 5 to 9 years regardless of AML subtype and sex.

### 3.5. Median of Net Survival in “Uncured” Patients by AML Subtype and Age

According to our assumption of cure testing, a large proportion of patients with a diagnosis of AML will not be “cured”. The MedS for “uncured” young patients (25 years) varied in men from less than 1 month for APL to 23 months (95% CI, 18–28) for AML-M2, and, in women, from 6 months (95% CI, 3–10) for t-AML to 21 months (95% CI, 16–26) for AML-M2. No major change in MedS was observed in the elderly (75 years old); MedS was less than 5 months regardless of AML subtype and sex (Figure 2D.1,D.2; Table 4).

According to sex, the largest difference in the MedS for “uncured” patients was observed in the youngest patients with t-AML, varying from 17 months in men to 6 months in women (Table 4).

## 4. Discussion

In this study, by using flexible modeling methods, we estimated the long-term net survival and proportion of cured patients according to AML subtype diagnosed during the 1995–2015 period in a French population-based cohort. AML survival and cure indicators varied according to AML subtype, age at diagnosis and sex.

Our results highlight various phenomena in AML. First, we confirmed that long-term survival is different according to AML subtype, with APL presenting the best prognosis. These results highlight progress in APL management with better knowledge of cytogenetic abnormalities and disease biology and the use of tailored treatments particularly ATRA/ATR as standard of care from the early 2000s [18,19].

The worst situation was observed in t-AML and AML-MRC. AML-NOS, probably because of its less-defined molecular profile, also presented poor prognosis. Our results for AML-NOS and AML-RCA are comparable to those observed in the Surveillance Epidemiology and End Results (SEER) database in the USA [20].

Taken together, t-AML, AML-MRC and AML-NOS represent 40% of AML cases, and thus, new clinical strategies need to be developed. Several clinical trials using new combination of chemotherapy report encouraging results in these entities [21,22].

A second point raised by our study is the comparatively poor survival of older patients. Even in AML-RCA, where specific treatment strategies are available, the 10-year NS did not exceed 11% in older patients (75 years). The presence of multiple comorbidities in the oldest patients could make access to more aggressive treatment difficult [23]. Nevertheless, special attention is warranted for such patients in France. A Swedish study has already shown encouraging results in elderly patients receiving a standard intensive treatment, with better short- and long-term survival compared to patients under palliative treatment [24].

More surprisingly, an atypical survival trend by age was found for AML-M4 and AML-M1 in both sexes and for AML-NOS (only for men), in which age had a non-linear effect on excess mortality, with a better prognosis for middle-aged than for both younger and older patients. We found also poor survival in younger patients with t-AML and AML-M0. Progress has been made in the outcome of childhood and young adult AML. While the AML outcome of infants (under the age of one year) was worse than that of older children, there have been improvements in recent years. Notably, the use of allogeneic hematopoietic stem cell transplantation in first complete remission has contributed to improved outcome of these particular patients compared to the oldest childhood patients [25]. However, age disparities remain and have been observed, with patients diagnosed between 10 and 19 years of age being at a higher risk of death compared to those diagnosed before age 10 (Hazard Ratio adjusted for AML subtype = 1.30 (95% CI, 1.17–1.44)) [26]. Further studies on the effects of age on outcome in pediatric AML are warranted to better define priority areas for future clinical research.

Our analysis highlights sex differences in the prognosis of younger AML patients. This appears to be in keeping with results from Hossain et al., who have already found a significant sex effect on the survival of pediatric and young adult AML patients in the US population, with a substantially increased risk of death in younger male patients diagnosed with APL and AML-inv (16) (20–24 years age) [27].

The most surprising results in our study were observed in younger patients with t-AML, where better prognosis was seen in men compared to women (5-year net survival equal to 40% (95% CI, 19–82) vs. 12% (95% CI, 3–53), respectively) and MedS of “uncured” patients from 17 months (95% CI, 10–25) in men to 6 months (95% CI, 3–10) in women, with P being similar in both sexes. A UK population-based study of patients diagnosed with AML in the period 2004–2013 showed that men with t-AML show better survival compared to women (5-year relative survival, 5.80 (95% CI, 1–17) in men and 1.2 (95% CI, 0–9.2) in women) [28]. In a US study, sex disparities in survival among adults with AML reduced over the 1973–2014 period [29]. The opposite was seen in patients diagnosed with t-AML at less than 25 years of age, where the 5-year all-cause survival ranged from 49.8% in women to 23.1% in men [30]. Molecular, genetic and other biological disease features, patient clinical profiles or lifestyle trajectories might explain this. Further studies will be required to understand the basis for survival differences between sexes according to AML subtypes, preferably in a larger study population. Indeed, younger patients were a minority of the AML patients in our analysis of the French FRANCIM database (large confidence interval for some AML subtypes).

The major finding of our study was the quantification of the number of AML patients that can be considered as cured. The marked decrease in the excess mortality rate observed 2 years after diagnosis, leading to relatively stable survival thereafter for the 10-year follow up period, allowed to accept the cure assumption in each AML subtype for a part of patients with different age groups. This could be demonstrated by using a flexible parametric tool that allows to keep the quantitative form of covariates while considering their complex effects. In the cure framework, several models were developed, but the flexible cure model proposed by Andersson et al. appears to be the most appropriate tool, especially in older cohorts [31]. Currently, the validation of the cure assumption is based on a graphical verification due to the lack of statistical tests, and the study of statistical cure needs a sufficiently long follow-up of patients.

In several AML subtypes, age groups concerned by cure are not the same according to sex. In AML-M0, one in two of the younger female patients are considered as cured (P = 44% in those aged 25), whereas the assumption of cure was not accepted in younger men. In patients aged 25 years with t-AML, P ranging from 10% to 31% in women and men, respectively. Even if few patients are diagnosed at an early age, few hypotheses could explain these sex differences.


Even if concerning a minority of AML patients, the finding that a proportion is considered as “cured” during their follow-up is of great importance. This is particularly true in the youngest patients, where it could help them to recover a normal life, at least regarding social status and access insurance and finance.

We observed an inverse relationship between the TTC and the proportion of “cured” patients: the highest P was found in patients with a shorter TTC. This important finding is likely to be related to changes in clinical practice relating to studies that report the importance of a rapid decrease of leukemic blasts in blood and bone marrow, as measured by flow cytometry, to reach complete remission, sustained control or clearance of minimal residual disease and consequentlong term event-free survival [32,33]. The low MedS of “uncured” patients, which did not exceed 23 months, reinforces the importance of achieving these response criteria.

Complete remission was obtained in 60% of AML cases, and of these, 59% relapsed with a median duration of first complete remission of 7.2 months. The median all-cause survival from relapse was 6 months [34]. The all-cause survival after relapse depended mainly on age and cytogenetics at diagnosis. Analysis of prognosis factors that influence the post-relapse survival outcome of childhood patients with AML found sex as a predictor, but it was not significant, with a higher risk of mortality observed in men compared to women (HR = 1.30, not significant) [35]. Sex differences in survival after relapsed AML (better for women) could be related to the slight sex differences seen in the time to cure, with a slightly reduced delay in women compared to men in several AML subtypes.

Studies conducted in different European countries have shown disparities in AML cure and survival of uncured patients. In Sweden, for example, an increase in the proportion of “cured” patients and a similar age disparity for cure as in our study were found, but the MedS in “uncured” patients was higher for younger patients than that in France [36]. By contrast, the same proportion of cure was found in younger patients between England and France, but the MedS of “uncured” patients is superior in France [37]. Geographical disparities in cure highlight differences in healthcare management between countries, particularly for younger patients, with probable differences in therapeutic guidelines and age recommendation to treat by stem cell transplantation and the proportion of patients enrolled in clinical trials. The direct link between treatment management and trends in epidemiological indicators needs to be investigated using specific datasets with detailed annotations regarding diagnostic practices and treatment.

In order to indirectly appreciate the impact of changing clinical practices on survival, we conducted a specific survival analysis by period of diagnosis, splitting the database into two periods of diagnosis: 1995–2005 and 2006–2015. This analysis strategy did not show significant differences in 5-year AML NS between the two study periods by either AML subtype or sex for all ages. A previous study based on population-based cancer registries in Switzerland between 2001 and 2013 found modestly improved survival over the period of diagnosis for all ages, with significantly improved outcomes only being seen in patients aged 65–74 years (5-year net survival from 5.2% to 13.5%; *p* < 0.001) [38]. These results are confirmed also in the SEER program study [39]. This good result observed in the age class of 65–74 years on a population-based level reflects recent progress made in the management of elderly patients, with increased use of allogeneic hematopoietic stem cell transplantation and access to clinical trials in this age class [38].

Our work allowed to identify AML subtypes where cure assumption is negative and/or settings where no progress has been achieved in AML cure or survival for uncured patients, thus pointing to priority areas for future research efforts. Mutational profiling has been integrated into AML classification [40] and in clinical decision making. Likewise, molecular MRD diagnostics is emerging as are rationalized treatments for long term control of residual disease in AML [41]. Future studies on the long-term survival of AML patients should be adjusted on these novel molecular and cellular characteristics which offer novel perspectives in future epidemiology studies.

## Figures and Tables

**Figure 1 jcm-10-01657-f001:**
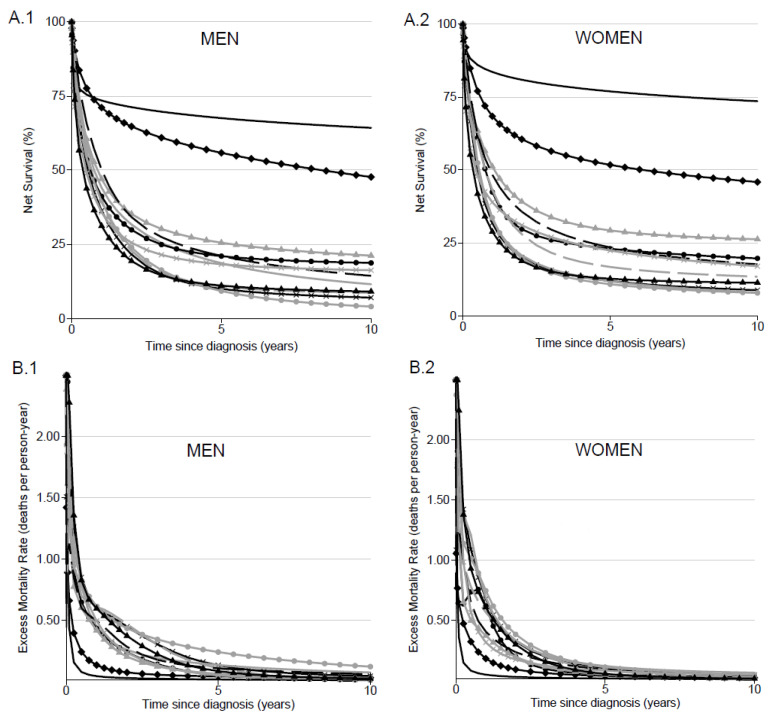
Net survival (NS) and excess mortality rate (EMR) for patients diagnosed from 1995 to 2015 according to AML subtype and sex: Net survival probabilities (%) over time since diagnosis in years in men (**A.1**) and in women (**A.2**); dynamics of the excess mortality rate (deaths per person-year) over the time since diagnosis in years in men (**B.1**) and in women (**B.2**); one-year net survival probabilities (%) over the age of diagnosis in men (**C.1**) and in women (**C.2**); five-year net survival probabilities (%) over the age of diagnosis in men (**D.1**) and in women (**D.2**).

**Figure 2 jcm-10-01657-f002:**
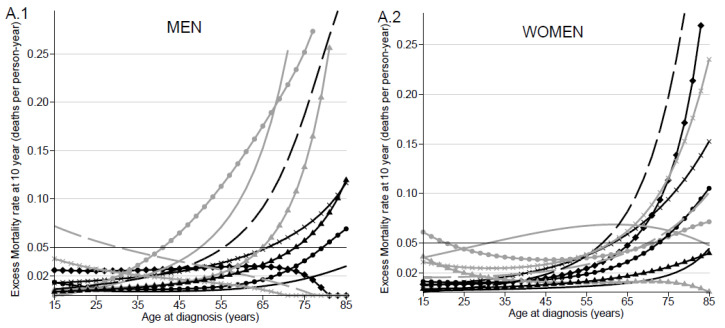
Excess mortality rate and cure indicators for patient diagnosed from 1995 to 2015 according to AML subtype and sex: Excess mortality rate (deaths per person-year) over the age at diagnosis in years in men (**A.1**) and in women (**A.2**); reference horizontal line at the excess mortality rate equal to 0.05 to fix the limit of the cure assumption accepted. Proportion of “cured” patients over the age at diagnosis in men (**B.1**) and in women (**B.2**). Time to cure over the age at diagnosis in men (**C.1**) and in women (**C.2**). Median survival time for “uncured” patients in months over the age at diagnosis in men (**D.1**) and in women (**D.2**).

**Table 1 jcm-10-01657-t001:** Classification of Acute Myeloid Leukaemia (AML) according to the Haemacare Study Group proposal as regards to their morphology (ICD-O-3 codes) and numbers of cases in our 9 453 AML patients over period of diagnosis from 1995 to 2015.

Specific Myeloid Morphologies	ICD-O-3 Morphology Codes	All	Men	Women
		N	N (%)	N (%)
All AML	-	9453	5001 (53)	4452 (47)
Acute promyelocytic leukemia with t(15;17) (q22;q12) (APL)	9866/3	575	310 (54)	265 (46)
AML with recurrent cytogenetic abnormalities (AML-RCA)	-	406	214 (53)	192 (47)
Acute myeloid leukemia with t(6;9)(p23;q34.1); DEK-NUP214	9865/3	6	3 (50)	3 (50)
Acute myeloid leukemia with inv(3)(q21.3q26.2) or t(3;3)(q21.3;q26.2); GATA2, MECOM	9869/3	6	3 (50)	3 (50)
Acute myeloid leukemia with inv(16)(p13.1q22) or t(16;16)(p13,1;q22); CBFB-MYH11	9871/3	173	86 (50)	87 (50)
Acute myeloid leukemia with t(8;21)(q22;q22); RUNX1-RUNX1T1	9896/3	135	82 (61)	53 (39)
Acute myeloid leukemia with t(9,11)(p22;q23); MLLT3-MLL	9897/3	84	39 (46)	45 (54)
Myeloid leukemia associated with Down syndrome	9898/3	2	1 (50)	1 (50)
AML with myelodysplasia-related changes (AML-MRC)	-	969	521 (54)	448 (46)
Acute Myeloid Leukemia with multilineage dysplasia	9895/3	903	483 (53)	420 (47)
Refractory anaemia with excess blasts in transformation	9984/3	66	38 (58)	28 (42)
Therapy-related AML; NOS (t-AML)	9920/3	613	300 (49)	313 (51)
Acute panmyelosis with myelofibrosis (APMF)	9931/3	85	63 (74)	22 (26)
Acute biphenotypic leukemia	-	109	66 (61)	43 (39)
Acute biphenotypic leukemia	9805/3	82	48 (59)	34 (41)
Mixed-phenotype acute leukemia with t(9;22)(q34.1;q11.2)	9806/3	5	2 (40)	3 (60)
Mixed-phenotype acute leukemia B/myeloid, NOS	9808/3	11	8 (73)	3 (27)
Mixed-phenotype acute leukemia T/myeloid, NOS	9809/3	11	8 (73)	3 (27)
Pure erythroid leukemia (AML-M6)	9840/3	279	180 (65)	99 (35)
Acute Myelomonocytic leukemia (AML-M4)	9867/3	844	439 (52)	405 (48)
Acute basophilic leukemia-	9870/3	3	2 (67)	1 (33)
Acute myeloid leukemia with minimal differentiation (AML-M0)	9872/3	372	221 (59)	151 (41)
Acute Myeloid leukemia without maturation (AML-M1)	9873/3	888	418 (47)	470 (53)
Acute Myeloid leukemia with maturation (AML-M2)	9874/3	1171	634 (54)	537 (46)
Acute monoblastic and monocytic leukemia (AML-M5)	9891/3	825	451 (55)	374 (45)
Acute megakarioblastic leukemia-	9910/3	63	39 (62)	24 (38)
Myeloid sarcoma-	9930/3	50	26 (52)	24 (48)
Acute Myeloid leukemia, not otherwise specified (AML-NOS)	9861/3	2201	1117 (53)	1084 (47)

**Table 2 jcm-10-01657-t002:** Characteristics of AML patients in the FRANCIM population-based study diagnosed on 1995–2015 by AML-subtype and sex: Numbers of cases, median Age, numbers of cases alive at 5, 10 years and deceased at 10 years (or on 30 June 2018) and % lost of follow-up (at 10 years or on 30 June 2018).

Specific AML Morphologies	Number at Diagnosis	Age at Diagnosis, Median (Min–Max)	Number at 5 Year	Number at 10 Year	Number Death at 10 Year	Percentage of Loss to Follow-up (%)
MEN						
Acute promyelocytic leukemia with t(15;17) (q22;q12) (APL)	310	57 (15–92)	146	66	123	2.2
AML with recurrent cytogenetic abnormalities (AML-RCA)	214	53 (15–91)	85	32	107	0
AML with myelodysplasia-related changes (AML-MRC)	521	74 (15–97)	39	13	482	<1
Therapy-related AML; NOS (t-AML)	300	72 (16–93)	15	3	274	<1
Acute panmyelosis with myelofibrosis (APMF)	63	72 (36–88)	6	2	58	1.6
Acute biphenotypic leukemia-	66	58 (15–85)	17	8	43	3
Pure erythroid leukemia (AML-M6)	180	68 (23–98)	21	10	151	1.1
Acute Myelomonocytic leukemia (AML-M4)	439	69 (15–93)	73	35	358	<1
Acute basophilic leukemia-	2	75 (70–80)	0	0	2	0
Acute myeloid leukemia with minimal differentiation (AML-M0)	221	72 (15–94)	16	6	201	<1
Acute Myeloid leukemia without maturation (AML-M1)	418	67 (18–93)	71	28	322	<1
Acute Myeloid leukemia with maturation (AML-M2)	634	70 (15–97)	98	39	526	<1
Acute monoblastic and monocytic leukemia (AML-M5)	451	67 (15–101)	54	21	379	0
Acute megakarioblastic leukemia -	39	64 (24–90)	2	1	36	2.6
Myeloid sarcoma -	26	58 (18–87)	6	2	20	0
Acute Myeloid leukemia, not otherwise specified (AML-NOS)	1117	73 (15–98)	103	64	1018	<1
WOMEN						
Acute promyelocytic leukemia with t(15;17) (q22;q12) (APL)	265	52 (15–96)	154	85	74	1.5
AML with recurrent cytogenetic abnormalities (AML-RCA)	192	53 (16–90)	81	35	99	2
AML with myelodysplasia-related changes (AML-MRC)	448	76 (15–98)	42	20	407	<1
Therapy-related AML; NOS (t-AML)	313	70 (15–93)	20	6	282	<1
Acute panmyelosis with myelofibrosis (APMF)	22	68 (39–88)	11	1	14	0
Acute biphenotypic leukemia-	43	60 (20–93)	7	0	32	0
Pure erythroid leukemia (AML-M6)	99	70 (19–98)	10	3	90	0
Acute Myelomonocytic leukemia (AML-M4)	405	68 (15–97)	68	32	318	1.8
Acute basophilic leukemia -	1	75 (75–75)	0	0	1	0
Acute myeloid leukemia with minimal differentiation (AML-M0)	151	72 (15–97)	20	5	128	0
Acute Myeloid leukemia without maturation (AML-M1)	470	68 (17–94)	107	43	343	<1
Acute Myeloid leukemia with maturation (AML-M2)	537	72 (15–103)	91	37	424	<1
Acute monoblastic and monocytic leukemia (AML-M5)	374	70 (15–96)	61	27	301	2.1
Acute megakarioblastic leukemia-	24	75 (25–85)	2	1	22	0
Myeloid sarcoma-	24	67 (19–92)	3	2	21	0
Acute Myeloid leukemia, not otherwise specified (AML-NOS)	1084	76 (15–102)	114	88	961	<1

**Table 3 jcm-10-01657-t003:** Estimated net survival in percent (and 95% confidence interval) at 1, 5 and 10 years after AML diagnosis according to AML subtype and sex for all ages and with age fixed at 25, 50 and 75 years old. Results of the flexible excess mortality model retained in each AML subtype and sex with the covariate of age at diagnosis fixed to values equal to 25, 50 and 75 years old.

% (95% CI)	1-Year Net Survival				5-Year Net Survival				10-Year Net Survival			
	All Age	Age = 25	Age = 50	Age = 75	All Age	Age = 25	Age = 50	Age = 75	All Age	Age = 25	Age = 50	Age = 75
MEN												
Acute promyelocytic leukemia with t(15;17) (q22;q12) (APL)	74 (69–79)	85 (77–94)	85 (79–91)	55 (45–67)	68 (63–74)	82 (74–91)	81 (74–88)	46 (35–59)	64 (58–71)	79 (70–89)	78 (70–86)	40 (28–56)
AML with recurrent cytogenetic abnormalities (AML-RCA)	72 (68–76)	96 (92–99)	84 (78–90)	25 (16–41)	56 (51–62)	86 (78–94)	66 (57–76)	10 (4–26)	48 (42–56)	75 (63–89)	56 (45–69)	9 (3–30)
AML with myelodysplasia-related changes (AML-MRC)	36 (33–40)	79 (67–94)	65 (58–72)	34 (30–39)	10 (8–13)	54 (34–85)	32 (24–41)	6 (4–9)	7 (5–10)	47 (28–82)	25 (18–35)	3 (2–6)
Therapy-related AML; NOS (t-AML)	40 (35–45)	67 (48–93)	49 (40–59)	39 (33–46)	9 (6–13)	40 (19–82)	16 (10–26)	7 (4–13)	4 (2–7)	36 (16–82)	9 (4–19)	2 (1–5)
Pure erythroid leukemia (AML-M6)	44 (38–51)	68 (43–100)	67 (57–77)	34 (26–44)	19 (14–25)	52 (24–100)	42 (32–56)	5 (2–13)	12 (7–18)	49 (22–100)	31 (20–47)	1 (0–9)
Acute Myelomonocytic leukemia (AML-M4)	41 (38–45)	72 (60–85)	73 (68–79)	22 (17–29)	21 (18–25)	44 (30–64)	49 (42–58)	4 (2–8)	19 (15–23)	40 (26–62)	45 (38–55)	3 (1–8)
Acute myeloid leukemia with minimal differentiation (AML-M0)	39 (34–46)	78 (62–97)	63 (54–74)	32 (25–41)	11 (7–16)	25 (9–69)	20 (12–32)	8 (4–14)	9 (5–16)	16 (3–74)	15 (7–29)	7 (3–17)
Acute Myeloid leukemia without maturation (AML-M1)	47 (43–51)	71 (58–85)	75 (69–81)	30 (25–38)	26 (22–29)	51 (37–70)	53 (46–62)	5 (3–10)	21 (17–26)	47 (33–68)	47 (39–57)	2 (1–7)
Acute Myeloid leukemia with maturation (AML-M2)	50 (47–54)	84 (77–93)	76 (72–81)	39 (34–44)	21 (18–24)	56 (44–73)	45 (39–53)	9 (6–13)	14 (12–18)	52 (38–70)	36 (29–45)	3 (2–7)
Acute monoblastic and monocytic leukemia (AML-M5)	36 (32–40)	72 (63–83)	59 (53–67)	20 (15–26)	18 (15–22)	44 (33–58)	34 (28–43)	7 (4–12)	16 (13–21)	36 (25–52)	30 (23–40)	7 (4–13)
Acute Myeloid leukemia, not otherwise specified (AML-NOS)	31 (29–34)	67 (58–77)	57 (52–62)	26 (23–30)	11 (9–13)	42 (32–55)	29 (24–34)	5 (3–7)	9 (7–11)	39 (29–53)	25 (20–30)	3 (2–5)
WOMEN												
Acute promyelocytic leukemia with t(15;17) (q22;q12) (APL)	84 (80–88)	95 (90–99)	89 (85–94)	70 (60–81)	77 (72–82)	92 (86–98)	84 (78–91)	58 (46–72)	74 (68–79)	90 (83–98)	81 (74–89)	52 (39–69)
AML with recurrent cytogenetic abnormalities (AML-RCA)	68 (64–73)	92 (86–99)	84 (78–91)	34 (24–49)	52 (47–58)	84 (74–94)	67 (58–78)	9 (4–20)	46 (40–53)	79 (68–92)	60 (49–73)	4 (1–18)
AML with myelodysplasia-related changes (AML-MRC)	32 (28–36)	87 (77–99)	67 (61–75)	29 (25–35)	12 (10–15)	73 (55–97)	41 (33–51)	6 (4–9)	9 (7–12)	68 (48–96)	33 (25–44)	3 (2–6)
Therapy-related AML; NOS (t-AML)	34 (29–39)	37 (19–74)	47 (39–57)	28 (23–36)	11 (8–15)	12 (3–53)	20 (14–30)	7 (4–12)	8 (5–13)	9 (2–49)	16 (10–26)	5 (2–10)
Pure erythroid leukemia (AML-M6)	33 (26–42)	84 (66–100)	58 (45–76)	24 (15–37)	12 (7–19)	57 (29–100)	25 (14–46)	6 (2–17)	8 (4–17)	43 (15–100)	16 (6–42)	3 (1–20)
Acute Myelomonocytic leukemia (AML-M4)	42 (39–47)	72 (61–85)	67 (61–74)	28 (22–35)	23 (19–27)	48 (35–67)	44 (37–53)	9 (6–14)	20 (16–24)	45 (31–65)	40 (32–49)	6 (3–12)
Acute myeloid leukemia with minimal differentiation (AML-M0)	43 (37–51)	76 (64–90)	65 (54–78)	33 (25–44)	17 (12–23)	48 (32–73)	33 (21–50)	6 (3–13)	13 (9–20)	42 (26–70)	27 (16–45)	3 (1–11)
Acute Myeloid leukemia without maturation (AML-M1)	51 (48–55)	85 (76–94)	81 (77–86)	36 (30–42)	29 (26–33)	56 (42–76)	58 (51–65)	12 (8–18)	26 (22–31)	49 (33–71)	53 (45–62)	11 (6–18)
Acute Myeloid leukemia with maturation (AML-M2)	48 (45–52)	84 (77–93)	78 (73–83)	41 (37–47)	23 (21–26)	68 (57–82)	54 (48–62)	9 (7–13)	18 (15–21)	66 (53–81)	46 (39–54)	3 (1–6)
Acute monoblastic and monocytic leukemia (AML-M5)	39 (35–43)	67 (54–82)	65 (58–73)	28 (22–35)	22 (19–26)	49 (36–67)	46 (38–55)	9 (6–14)	17 (14–21)	42 (27–63)	37 (29–48)	4 (2–9)
Acute Myeloid leukemia, not otherwise specified (AML-NOS)	29 (27–31)	81 (74–88)	62 (57–66)	23 (20–26)	13 (11–14)	62 (52–73)	35 (30–41)	5 (3–7)	11 (10–13)	59 (49–70)	32 (27–38)	4 (2–6)

**Table 4 jcm-10-01657-t004:** Estimated Proportion of cured patient in percent, Time to cure in years and Median survival for ‘uncured’ patient (and 95% Confidence Interval) when cure assumption was acceptable, according to AML-subtype and sex for all age and with age fixed at 25, 50 and 75 years old. Results of the flexible cure model retained in each AML-subtype and sex with the covariate age at diagnosis fixed to the value equal to 25, 50 and 75 years old.

	Age Class with Cure Assumption Accepted	Proportion of Cured Patient (%)			Time to Cure (Years)			Median Survival for Uncured Patient (Months)		
		Age = 25	Age = 50	Age = 75	Age = 25	Age = 50	Age = 75	Age = 25	Age = 50	Age = 75
MEN										
Acute promyelocytic leukemia with t(15;17) (q22;q12) (APL)	15–85	79 (68–87)	78 (69–85)	40 (28–52)	2 (0–8)	3 (0–7)	6 (0–15)	0 (0–0)	1 (0–1)	1 (0–2)
AML with recurrent cytogenetic abnormalities (AML-RCA)	15–85	79 (66–88)	61 (49–71)	4 (1–12)	4 (0–9)	6 (0–12)	9 (0–39)	19 (11–27)	17 (10–23)	4 (2–7)
AML with myelodysplasia-related changes (AML-MRC)	15–66	47 (22–70)	25 (18–34)	no cure	6 (0–20)	7 (0–15)	no cure	18 (12–24)	13 (10–17)	no cure
Therapy-related AML; NOS (t-AML)	15–42	31 (8–58)	no cure	no cure	7 (0–29)	no cure	no cure	17 (10–25)	no cure	no cure
Pure erythroid leukemia (AML-M6)	15–50	43 (8–75)	33 (21–46)	no cure	7 (0–35)	7 (0–18)	no cure	15 (5–25)	14 (8–20)	no cure
Acute Myelomonocytic leukemia (AML-M4)	15–80	41 (24–57)	46 (38–54)	3 (1–6)	6 (0–18)	5 (0–11)	7 (0–25)	13 (9–17)	12 (9–15)	3 (2–4)
Acute myeloid leukemia with minimal differentiation (AML-M0)	30–85	no cure	17 (9–27)	5 (2–11)	no cure	6 (0–20)	7 (0–26)	no cure	14 (10-17)	4 (3–6)
Acute Myeloid leukemia without maturation (AML-M1)	15–66	45 (28–61)	47 (38–56)	no cure	6 (0–18)	6 (0–12)	no cure	13 (8–18)	13 (10–17)	no cure
Acute Myeloid leukemia with maturation (AML-M2)	15–56	48 (32–62)	35 (28–43)	no cure	8 (0–17)	8 (3–13)	no cure	23 (18–28)	18 (15–21)	no cure
Acute monoblastic and monocytic leukemia (AML-M5)	15–85	41 (28–53)	32 (24–40)	4 (2–8)	5 (0–14)	6 (0–12)	7 (0–21)	10 (8–13)	8 (6–10)	3 (2–4)
Acute Myeloid leukemia, not otherwise specified (AML-NOS)	15–72	38 (27–49)	25 (20–30)	no cure	6 (0–15)	6 (1–12)	no cure	11 (9–14)	9 (7–11)	no cure
WOMEN										
Acute promyelocytic leukemia with t(15;17)(q22;q12) (APL)	15–85	90 (80–95)	82 (73–88)	55 (41–68)	1 (0–2)	2 (0–5	5 (0–16)	7 (0–14)	6 (0–13)	4 (0–9)
AML with recurrent cytogenetic abnormalities (AML-RCA)	15–65	79 (65–88)	62 (50–72)	no cure	4 (0–9)	6 (0–12)	no cure	20 (12–28)	19 (12–25)	no cure
AML with myelodysplasia-related changes (AML-MRC)	15–62	68 (39–85)	33 (24–43)	no cure	5 (0–15)	7 (0–14)	no cure	17 (11–23)	12 (9–16)	no cure
Therapy-related AML; NOS (t-AML)	20–72	10 (1–31)	17 (10–25)	no cure	7 (0–46)	7 (0–18)	no cure	6 (3–10)	8 (6–10)	no cure
Pure erythroid leukemia (AML-M6)	15–30	51 (10–82)	no cure	no cure	6 (0–36)	no cure	no cure	15 (7–22)	no cure	no cure
Acute Myelomonocytic leukemia (AML-M4)	15–72	45 (28–60)	40 (32–48)	no cure	6 (0–17)	6 (0–12)	no cure	12 (10–15)	11 (9–12)	no cure
Acute myeloid leukemia with minimal differentiation (AML-M0)	15–70	44 (23–63)	29 (16–44)	no cure	6 (0–20)	6 (0–20)	no cure	14 (8–20)	12 (8–17)	no cure
Acute Myeloid leukemia without maturation (AML-M1)	15–85	51 (33–67)	55 (46–62)	9 (6–14)	6 (0–17)	5 (0–10)	8 (0–19)	17 (13–21)	15 (12–18)	5 (4–7)
Acute Myeloid leukaemia with maturation (AML-M2)	15–58	64 (48–76)	47 (39–54)	no cure	5 (0–12)	6 (2–10)	no cure	21 (16–26)	17 (14–21)	no cure
Acute monoblastic and monocytic leukemia (AML-M5)	15–60	45 (29–60)	41 (32–50)	no cure	4 (0–13)	4 (0–9)	no cure	8 (5–12)	9 (6–11)	no cure
Acute Myeloid leukemia, not otherwise specified (AML-NOS)	15–85	58 (47–68)	32 (26–37)	4 (2–5)	5 (0–10)	6 (1–10)	7 (0–16)	13 (11–16)	10 (8–11)	3 (3–4)

## Data Availability

No new data were created or analyzed in this study. Data sharing is not applicable to this article.

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
