# Peer review of "Flexible Modeling of Net Survival and Cure by AML Subtype and Age: A French Population-Based Study from FRANCIM"

_jcm, 2021, doi:10.3390/jcm10081657_

Round 1

Reviewer 1 Report

Mounier et al. submitted a manuscript entitled "Flexible modelling of net survival and cure by AML subtype and age; a French population-based study
from FRANCIM". The manuscript is well written and very clear. The main limit for this study is the time period between 1995 and 2015. We can't conclude an analysis of survival with not considering the evolution of the treatment during this period: antifungal, improvement of AlloSCT in elderly patients, higher dose of daunorubicin, hypomethylating agents ...etc. I recommend to split the analysis in several periods: 1995-2005, 2005-2010, 2010-2015 for example. It could be interesting to use their model but considering the improvement of the treatment. 

Reviewer 2 Report

The proposed manuscript reports the results of a French population based study on AML. Its objectives were to study survival and cure in a cohort of more than 9000 cases diagnosed over a 20-year period (1995-2015).

The results highlight the heterogeneity of the disease with differences of net survival and according to age or disease categories. this allows for the identification of sub-groups with high unmet medical needs. 

One of the major issues is the relevance of the disease classification used for the study. this classification is not prognostic and some classes, such as AML-RCA encompass groups with excellent (APL), good (CBF leukemias) or very bad prognosis (inv(3)). This flaws the conclusion of the study in this specific group. In the study, approximate 50% of the patients in the AML-RCA class correspond to acute promyelocytic leukemia which has very specific evolution, treatment and outcome and this should be highlighted and discussed. On the other hand, it is well known that the FAB categories are associated with various molecular abnormalities (such as NPM1, FLT3) which have a clear impact on treatment response. 

The impact of sex on patient outcome is also deeply analyzed and discussed. The authors should however explain why is sex not identified as a pronostic variable in most AML studies. 

The discussion is confusing with a long list of results presented by age, sex or AML type without logic (last paragraph page 12). The message is unclear

Minor comments:

  • the poor NS of the patients aged 15-25 in several AML categories is an important observation and should be discussed in more details
  • what is the meaning of age categories such as "age 25" " age 50" etc..
  • page 13: the statement that disparities between EU countries in terms of access to cytogenetics does not correspond to the reality. The authors should delete this part or give references. 

Reviewer 3 Report

Dear Editor-in-chief,

this paper entitled “Flexible modelling of net survival and cure by AML subtype and age; a French population-based study from FRANCIM” is an original article reporting the results of a computational analysis from the French database FRANCIM. Authors used a flexible model to estimate net survival, the time to cure and the median survival of ‘uncured’ patients by AML subtype and age. The data are certainly of interest and the paper is clear and well written. I have just few minor comments to the authors:

  • The quality of the tables (particularly Table 3 and 4) and figures is low, and they are difficult to read. I suggest the authors reworking them to make it easier to read.
  • Authors in the text comments about 10 y net survival and the data are presented only in Table 3. I would suggest adding a graph with this data.
  • I would recommend the authors to specify in the caption of Figure 2 what the horizontal lines in A.1 and A.2 represent.
  • Authors in the Discussion comment upon the differences of MedS in ‘uncured’ patients between the two sex that is particularly evident for young adults. This difference seems to resemble the sex disparities present in pediatric relapsed AML as shown in Dini et al., Blood, 2011. I would suggest the authors further comment on this interesting finding.
  • Authors state that one of the goals of this paper is to identify areas of future research. To this aim it would be interesting to compare results between patients treated from 1995-2005 and patients treated from 2005 and 2015. This may give some interesting information on the area in which improvements have been done.
  • I would suggest the author considering adding in the conclusions the necessity to extend this type of studies also in pediatric patients considering the differences in outcomes between different ages as shown for example in Masetti et al., Haematologica, 2014.

Round 2

Reviewer 1 Report

The authors answered to all my comments. 

Reviewer 2 Report

no comments